# CIRCUIT REPRESENTATION LEARNING WITH MASKED GATE MODELING AND VERILOG-AIG ALIGNMENT

**Haoyuan Wu**[1*]  **Haisheng Zheng**[2*]  **Yuan Pu**[1,3]  **Bei Yu**[1]
[1]The Chinese University of Hong Kong
[2]Shanghai Artificial Intelligence Laboratory   [3]ChatEDA Tech
{hywu24,ypu,byu}@cse.cuhk.edu.hk   zhenghaisheng@pjlab.org.cn

## ABSTRACT

Understanding the structure and function of circuits is crucial for electronic design automation (EDA). Circuits can be formulated as And-Inverter graphs (AIGs), enabling efficient implementation of representation learning through graph neural networks (GNNs). Masked modeling paradigms have been proven effective in graph representation learning. However, masking augmentation to original circuits will destroy their logical equivalence, which is unsuitable for circuit representation learning. Moreover, existing masked modeling paradigms often prioritize structural information at the expense of abstract information such as circuit function. To address these limitations, we introduce MGVGA, a novel constrained masked modeling paradigm incorporating masked gate modeling (MGM) and Verilog-AIG alignment (VGA). Specifically, MGM preserves logical equivalence by masking gates in the latent space rather than in the original circuits, subsequently reconstructing the attributes of these masked gates. Meanwhile, large language models (LLMs) have demonstrated an excellent understanding of the Verilog code functionality. Building upon this capability, VGA performs masking operations on original circuits and reconstructs masked gates under the constraints of equivalent Verilog codes, enabling GNNs to learn circuit functions from LLMs. We evaluate MGVGA on various logic synthesis tasks for EDA and show the superior performance of MGVGA compared to previous state-of-the-art methods. Our code is available at https://github.com/wuhy68/MGVGA.

## 1 INTRODUCTION

In recent years, there has been a surge of interest in deep learning for electronic design automation (EDA), which holds great potential for achieving faster design closure and minimizing the need for extensive human supervision (Wen et al., 2022; Chen et al., 2022; Liang et al., 2023; Chen et al., 2023; Wu et al., 2024). Logic synthesis (Hachtel & Somenzi, 2005), a vital step in EDA, is a process by which an abstract specification of desired circuit behavior is turned into a design implementation for logic gates. In the field of logic synthesis, circuits can be formulated as graphs (e.g., And-Inverter graph (AIG) (Mishchenko et al., 2006)), which are well-suited for modeling element connections and topology. Consequently, GNNs (Zhang et al., 2020; Zheng et al., 2024; Chowdhury et al., 2022) have been widely used to learn the characteristics of circuits for various downstream tasks.

The effectiveness of GNNs in logic synthesis has been demonstrated in supervised settings with task-specific labels. However, obtaining labeled data for supervised learning is costly while unlabeled circuit data is available and abundant. This discrepancy makes self-supervised learning suitable for circuit representation learning. Recent works (Wang et al., 2022; Shi et al., 2023) explored leveraging the functional aspects of circuits, such as truth tables and functional equivalence, to derive meaningful representations via the self-supervised learning paradigm. These methods efficiently capture the functional behaviors of circuits, which are crucial for many applications.

The structure of a circuit, including its layout, connectivity, gate numbers, and circuit level, plays a critical role in determining its power, performance, and area (PPA), all of which are key optimization targets of EDA. Models trained with functional targets often fall short in extracting structural

---

*These authors contributed equally to this work.

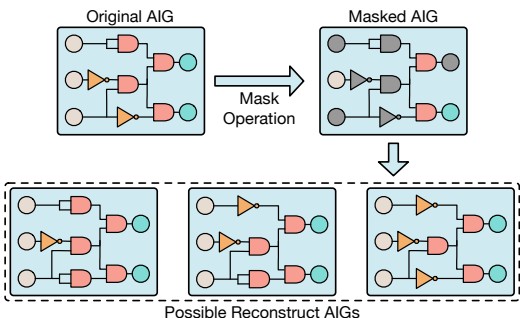

Figure 1: Possible reconstruct AIGs for masked AIG. If circuit gates are masked, there are various logic-correct solutions for reconstruction.

Figure 2: Logic equivalence between Verilog code and AIG. For a circuit design, AIG can be translated from Verilog code.

details. Masked modeling paradigms, which have been successfully applied in computer vision (He et al., 2022; Bao et al., 2021), natural language processing (Kenton & Toutanova, 2019), and graph learning (Hou et al., 2023; Li et al., 2023a), offer a promising solution to learn detailed structural information. Consequently, we apply the masked modeling paradigm to circuit representation learning to extract a more fine-grained representation of circuit structure.

Challengingly, traditional masked graph modeling paradigm (Hou et al., 2023; Li et al., 2023a) can not be applied directly to circuit representation learning which follows strict logical equivalence. In conventional applications, such as social or molecular graphs, masking nodes can provide a unique solution for reconstruction. However, when gates are masked in a circuit, their reconstruction will admit various solutions as illustrated in Figure 1. This is because no matter how we replace gates in the original circuits, logical correctness can still be maintained without necessarily preserving logical equivalence. Consequently, applying traditional masked modeling to circuit representation learning can not guarantee the extraction of circuit-specific features. To address this limitation, we propose a constrained masked modeling paradigm that ensures logical equivalence between the original and reconstructed circuits, thereby enabling effective circuit representation learning.

As mentioned, abstract circuit functions are useful for EDA tasks. However, they are not explicitly represented in the structural layouts. Such functional attributes are often derived from textual descriptions in hardware description languages (HDLs), which large language models (LLMs) can effectively process. LLMs have demonstrated remarkable performance in HDL code generation Pei et al. (2024); Tsai et al. (2024); Fang et al. (2024); Liu et al. (2023). Consequently, LLMs can guide GNNs in understanding circuit functions. Specifically, the AIG is logically equivalent to the corresponding behavior Verilog code for the same circuit design as illustrated in Figure 2. This equivalence allows for the alignment between Verilog codes and AIGs, facilitating GNNs' understanding of circuit functions.

This study presents MGVGA, a constrained masked modeling paradigm incorporating Masked Gate Modeling (MGM) and Verilog-AIG Alignment (VGA) for circuit representation learning. Firstly, we introduce MGM to mask gates in the latent space instead of masking in the original circuits. This approach can use unmasked gate representations as constraints to maintain logical equivalence during masked gate reconstruction, as these representations already capture features from masked gates. However, GNNs trained with MGM alone focus primarily on structural relationships within circuits, potentially overlooking function features. To address this, we design VGA to leverage LLMs' expertise in Verilog code to transfer functional knowledge to GNNs by aligning AIGs with equivalent Verilog codes. Specifically, VGA performs a masking operation on the original circuits and reconstructs masked gates under the constraint of corresponding Verilog codes. Although this operation will destroy the logical equivalence of the original circuits when cooperating with the traditional masked modeling strategy, we can still utilize equivalent Verilog codes as constraints to ensure the logic equivalence during the reconstruction process.

In summary, our main contributions are as follows:

- Propose masked gate modeling (MGM) for circuit representation learning, enabling GNNs to extract circuit representations with fine-grained structural information.

- Develop the Verilog-AIG alignment (VGA), which employs LLMs as teachers to guide GNNs to extract circuit representations with abstract circuit functions through equivalent AIGs and Verilog codes alignment.

- Conduct extensive evaluations and show superior performance on various logic synthesis tasks including quality of result (QoR) and logic equivalence identification compared to previous state-of-the-art (SOTA) methods.

## 2 RELATED WORK

**AIG-Formatted Circuit Representations for Logic Synthesis.** An AIG is a directed acyclic graph (DAG) utilized for representing circuits (Brummayer & Biere, 2006), which is composed of AND gate, NOT gate, and terminal nodes that serve as primary inputs (PIs) and primary outputs (POs). Considering its simplicity, AIG-formatted circuits are widely used to perform circuit representation learning via GNNs. For example, Zhang et al. (2020) employs a GNN model to predict a circuit's power consumption, and Zheng et al. (2024) proposes a customized GNN to predict the delay of circuits accurately. Moreover, Wang et al. (2022) and Shi et al. (2023) perform self-supervised learning for extracting general AIG-formatted circuit representation. In this study, we convert circuits to AIGs to perform general circuit representation learning via a self-supervised learning paradigm.

**Masked Graph Autoencoder.** Masked autoencoders (He et al., 2022; Bao et al., 2021) are grounded in the masked modeling learning paradigm, which involves masking a portion of the input signals and predicting the obscured content. Graph autoencoders (Hou et al., 2023; Li et al., 2023a) employ an autoencoder architecture to encode nodes into latent representations and reconstruct the graph from these embeddings. Integrating the strengths of the above methods, the masked graph autoencoder has been introduced to advance representation learning for graph-structured data. The masked graph autoencoder randomly masks a subset of graph nodes and reconstructs them using information from the unmasked nodes and their structural connections. This approach compels the encoder to decipher the underlying relational patterns within the graph, thereby generating robust and informative node representations. In this paper, we apply the constrained masked modeling paradigm for general circuit representation learning through the masked graph autoencoder.

**LLM-based Embedding Models.** LLMs, structured as decoder-only architectures, inherently face challenges in effectively encoding bidirectional context, which can impede their capacity to generate comprehensive and discriminative embeddings. Consequently, researchers apply contrastive learning for representation learning of LLMs to leverage the natural language comprehension capabilities of LLMs for embedding-related tasks (Muennighoff et al., 2024; BehnamGhader et al., 2024; Li et al., 2023b; Lee et al., 2024). This kind of strategy can help LLMs extract better representation through bidirectional attention mechanisms without hurting their abilities. In this study, we utilize LLMs to extract Verilog codes embedding with a comprehensive understanding of circuit function, serving as constraints of the reconstruction process of VGA.

## 3 METHODOLOGY

Due to logical equivalence issues, traditional mask graph modeling techniques are inadequate for circuit representation learning. Additionally, GNNs have inherent limitations in extracting abstract circuit functions. To address these challenges, we propose MGVGA, a novel constrained masked modeling strategy incorporating masked gate modeling (MGM) and Verilog-AIG alignment (VGA) for enhanced circuit representation learning. AIGs have gained widespread adoption in circuit representation learning. Consequently, we convert circuits to AIGs to implement our MGVGA. In the following subsections, we will elucidate the details of MGM (Section 3.2) and VGA (Section 3.3) utilizing the AIG autoencoder. Notably, we also provide a detailed illustration of how we preserve the logic equivalence when performing MGVGA in Appendix A.1.1.

### 3.1 AIG AUTOENCODER

Let $\mathcal{G} = (\mathcal{V}, \mathcal{A})$ represent an AIG, where $\mathcal{V}$ denotes the set of $N$ nodes, $v_i \in \mathcal{V}$, categorized into four types: PI, PO, AND, and NOT gates, each labeled by $c_i \in \mathcal{C}, i \in \{1, 2, 3, 4\}$. The adjacency matrix

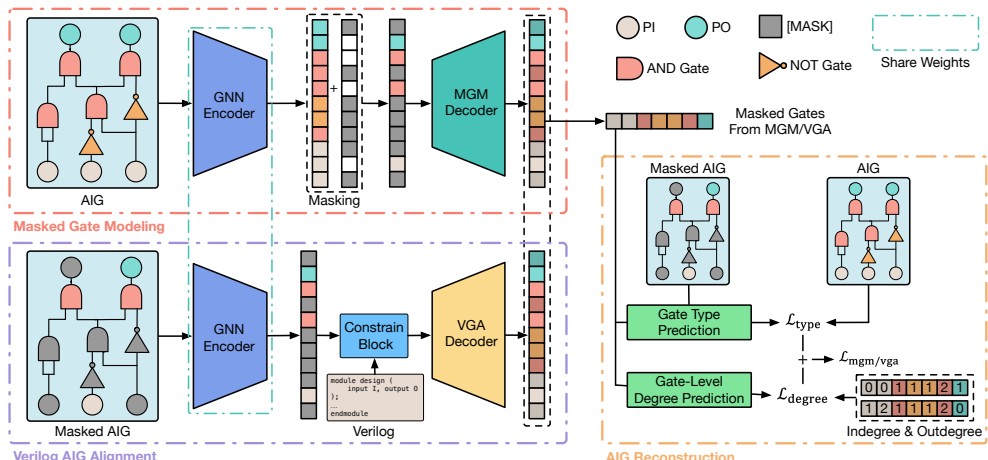

Figure 3: Overview of the MGVGA for circuit representation including masked gate modeling and Verilog-AIG alignment. For both MGM and VGA, the AIG reconstruction is implemented by gate type prediction and gate-level degree prediction from reconstructed representation.

$\mathcal{A} \in \{0,1\}^{N \times N}$ shows the connectivity between nodes, where $\mathcal{A}_{i,j} = 1$ represents an existing edge from $v_i$ to $v_j$. $\mathcal{A}$ delineates the structure and the types of connections within $\mathcal{G}$.

For an AIG autoencoder, a GNN encoder, denoted by $g_E$, encodes $\mathcal{G}$ into a latent space representation $\mathbf{X} \in \mathbb{R}^{N \times d}$, where $d$ represents the dimension of this representation. The encoding process of an AIG can be formulated as:

$$\mathbf{X} = g_E(\mathcal{V}, \mathcal{A}). \quad (1)$$

Concurrently, a GNN decoder, $g_D$, endeavors to reconstruct the AIG $\mathcal{G}$ from $\mathbf{X}$ according to:

$$(\tilde{\mathbf{X}}, \mathcal{A}) = \tilde{\mathcal{G}} = g_D(\mathbf{X}, \mathcal{A}), \quad (2)$$

where $\tilde{\mathcal{G}}$ denotes the reconstructed graph, potentially encompassing both node features and structure. The primary objective of the AIG autoencoder is to optimize the encoder $g_E$ to produce an effective representation $X$ that facilitates the accurate reconstruction of the original $\mathcal{G}$. Notably, following previous masked modeling paradigms (Hou et al., 2022; 2023), we do not mask the adjacent matrices $\mathcal{A}$ during the entire process and the detailed reasons will be explained in Appendix A.1.2.

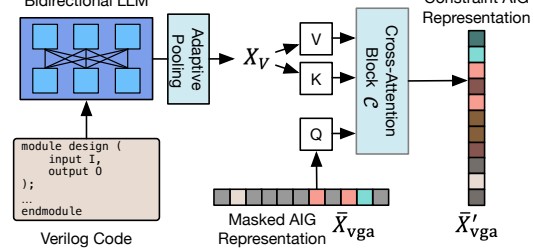

Figure 4: The constraint block for VGA.

## 3.2 MASKED GATE MODELING

The idea of the masked autoencoder has been applied successfully to graph self-supervised learning. As an extension of denoising autoencoders, masked graph autoencoder (Hou et al., 2023; Li et al., 2023a) selectively obscure portions of the graph, such as node features or edges, through a masking operation, and learn to predict the obscured content. In traditional masked graph autoencoders, nodes are typically masked directly in the input graph before being processed through the autoencoder framework. However, this approach presents significant challenges when applied to structurally constrained graphs like AIGs, which follow strict logical rules. Utilizing a straightforward random masking technique will lead to reconstructed logic expressions that diverge from their original forms, which can not be tolerated.

To address this, we propose masked gate modeling, where the AIG is initially processed unmasked through the encoder to capture its latent representation. Rather than masking nodes at the original AIG, the masking operation is applied to the encoded representation in the latent space. During the masked modeling process, the encoder can retain the complete logical structure of the AIG before masking. This approach allows the representations of unmasked gates to serve as constraints for reconstructing the attributes of masked gates, as the latent representations of unmasked gates have already aggregated some features from masked gates.

Formally, as depicted in Figure 3, we uniformly sample a subset of gates $\mathcal{V}_{\mathrm{mgm}} \subset \mathcal{V}$ without replacement and replace the remaining nodes with the mask token [MASK], which can be represented by a learnable vector $\mathbf{m} \in \mathbb{R}^d$. Consequently, the masked node representation $\bar{\mathbf{x}}_i \in \bar{\mathbf{X}}_{\mathrm{mgm}}$ for each node $v_i$ is given by:

$$\bar{\mathbf{x}}_i = \begin{cases} \mathbf{x}_i, & \text{if } v_i \in \mathcal{V}_{\mathrm{mgm}}; \\ \mathbf{m}, & \text{if } v_i \notin \mathcal{V}_{\mathrm{mgm}}. \end{cases} \tag{3}$$

The $\bar{\mathbf{X}}_{\mathrm{mgm}}$ is then fed into the decoder $g_D$ to reconstruct the $\mathcal{G}$ following Equation (2). As illustrated in Figure 3, the decoder maintains the connectivity of each node and generates the reconstructed node representation $\tilde{\mathbf{X}}_{\mathrm{mgm}} \in \mathbb{R}^{N \times d}$ for $\mathcal{G}$ reconstruction (Section 3.4).

## 3.3 VERILOG-AIG ALIGNMENT

Although GNNs enhanced with MGM excel at extracting structural information from circuits, they often struggle to capture abstract circuit functions that are not explicitly represented in structural layouts. Meanwhile, Verilog codes contain substantial semantic information, including high-level abstract concepts and functional logic in circuit designs. Recent studies (Lu et al., 2024; Pei et al., 2024) have begun leveraging LLMs to analyze Verilog codes, highlighting the potential for distilling circuit function knowledge from LLMs to GNNs. LLMs can serve as excellent teachers, guiding GNNs in understanding circuit functions through the alignment process of equivalent Verilog codes and AIGs. Meanwhile, as illustrated in Figure 2, AIGs are translated from Verilog codes via logic synthesis tools. Consequently, there exist equivalent behavior-level Verilog codes for AIGs. Based on equivalent Verilog-AIG pairs, we can perform Verilog-AIG alignment, which conducts masking operations on original AIGs. Subsequently, the masked gates are reconstructed under the constraints of equivalent Verilog codes. This equivalence allows for the reconstruction of masked AIGs under the supervision of Verilog code, facilitating GNNs' understanding of circuit functions.

Similar to MGM, we uniformly sample a subset of gates $\mathcal{V}_{\mathrm{vga}} \subset \mathcal{V}$ without replacement and replace the node types of remaining nodes with $c_m$, which represents these nodes are masked in the original AIG. Consequently, for $v_i \in \bar{\mathcal{V}}$ of the masked AIG, the node type $c_i$ can be defined as:

$$c_i = \begin{cases} c_i, & \text{if } v_i \in \mathcal{V}_{\mathrm{vga}}; \\ c_m, & \text{if } v_i \notin \mathcal{V}_{\mathrm{vga}}. \end{cases} \tag{4}$$

The masked AIG $\bar{\mathcal{G}} = (\bar{\mathcal{V}}, \mathcal{A})$ is fed into the encoder $g_E$ to generate the encoded masked AIG representation $\bar{\mathbf{X}}_{\mathrm{vga}}$ following Equation (1).

As mentioned earlier, the reconstruction of $\mathcal{G}$ from $\bar{\mathbf{X}}_{\mathrm{vga}}$ must be constrained by equivalent Verilog code to ensure strict logical equivalence. Consequently, we design a constraint block inspired by Jaegle et al. (2021) and Lee et al. (2024) as illustrated in Figure 4. Specifically, we perform adaptive pooling on token embeddings of Verilog code generated by LLMs to extract $\mathbf{X}_V \in \mathbb{R}^{M \times d_v}$, the representation of the equivalent Verilog code. We then feed the masked AIG representation $\bar{\mathbf{X}}_{\mathrm{vga}}$ and Verilog code representation $\mathbf{X}_V$ into a cross-attention block $\mathcal{C}$ to perform alignment between the masked AIG and Verilog code, with $\bar{\mathbf{X}}_{\mathrm{vga}}$ being projected to query $Q \in \mathbb{R}^{N \times d}$ and $\mathbf{X}_V$ being projected to key $K \in \mathbb{R}^{M \times d}$ and value $V \in \mathbb{R}^{M \times d}$. Specifically, we selected $M = 16$ after carefully balancing computational cost and performance. The output of the cross attention block (Vaswani, 2017) is the constrained AIG representation $\bar{\mathbf{X}}'_{\mathrm{vga}} \in \mathbb{R}^{N \times d}$, which can be calculated as follows:

$$\bar{\mathbf{X}}'_{\mathrm{vga}} = \mathcal{C}(\bar{\mathbf{X}}_{\mathrm{vga}}, \mathbf{X}_V) = \mathrm{Softmax}\left(\frac{QK^T}{\sqrt{d}}\right) V. \tag{5}$$

After aligning the logically equivalent Verilog code and masked AIG, $\bar{\mathbf{X}}'_{\mathrm{vga}}$ incorporates information from the abstract circuit function extracted by LLMs. Then, as illustrated in Figure 3, we obtain the reconstructed circuit representation $\tilde{\mathbf{X}}_{\mathrm{vga}}$ from $\bar{\mathbf{X}}'_{\mathrm{vga}}$ via $g_D$ following Equation (2) while preserving logical equivalence. Subsequently, $\tilde{\mathbf{X}}_{\mathrm{vga}}$ will be utilized for $\mathcal{G}$ reconstruction, detailed in Section 3.4.

## 3.4 AIG RECONSTRUCTION

As illustrated in Figure 3, given the reconstructed node representations from MGM or VGA, we can predict the attributes of masked nodes. First, we predict the types of each masked node, categorizing them as AND gate, NOT gate, PI, or POs. Next, we focus on the specific attributes of the nodes

themselves. Specifically, AND gates have two inputs (which may be identical), while NOT gates have only one input. Moreover, PIs have no inputs, and POs have no outputs. Consequently, we predict the degree of each masked node to help GNNs learn the attributes of each gate more effectively. Moreover, gate-level degree prediction aids GNNs in capturing the connectivity between gates and the overall structures of AIGs, as illustrated in Figure 3. For clarity, we unify the reconstructed node representations from MGM ($\tilde{\mathbf{X}}_{\text{mgm}} \in \mathbb{R}^{N \times d}$) and VGA ($\tilde{\mathbf{X}}_{\text{vga}} \in \mathbb{R}^{N \times d}$) into a single notation $\tilde{\mathbf{X}}$.

**Gate Type Prediction.** For gate type prediction, $\tilde{\mathbf{X}}$ is transformed by a mapping function $f_{\text{type}}$ : $\mathbb{R}^d \to \mathbb{R}^C$ into a categorical probability distribution over $C$ classes. This leads to the formation of the overall probability distribution matrix $\tilde{\mathbf{Z}} \in \mathbb{R}^{N \times C}$, where each element $\tilde{\mathbf{Z}}_{i,j}$ represents the softmax-estimated probability that node $v_i$ belongs to class $c_j$. Importantly, the gate type reconstruction loss is calculated only for the $N_m$ masked nodes:

$$\mathcal{L}_{\text{type}} = -\frac{1}{N_m} \sum_{\substack{i=1 \\ v_i \notin \mathcal{V}_u}}^{N} \sum_{j=1}^{C} \mathbf{Y}_{i,j} \log \tilde{\mathbf{Z}}_{i,j}, \tag{6}$$

where $\mathbf{Y} \subset \{0,1\}^{N \times C}$ and $\mathbf{Y}_{ij}$ is a binary indicator that equals 1 if the node $v_i$ belongs to class $c_j$ and 0 otherwise.

**Gate-Level Degree Prediction.** The gate-level degree prediction involves forecasting the in-degree and out-degree of each masked gate within the AIG. Formally, given the reconstructed node representations $\tilde{\mathbf{X}}$, in-degree labels $\mathbf{D}^- \in \mathbb{R}^N$, and out-degree labels $\mathbf{D}^+ \in \mathbb{R}^N$, we utilize mean squared error as the loss function for degree regression tasks. The degree reconstruction loss, calculated only for the $N_m$ masked nodes, is defined as:

$$\mathcal{L}_{\text{degree}} = \frac{1}{N_m} \sum_{\substack{i=1 \\ v_i \notin \mathcal{V}_u}}^{N} \left( (\mathbf{D}_i^- - f_{\text{in}}(\tilde{\mathbf{X}}_i))^2 + (\mathbf{D}_i^+ - f_{\text{out}}(\tilde{\mathbf{X}}_i))^2 \right), \tag{7}$$

where $f_{\text{in}} : \mathbb{R}^d \to \mathbb{R}$ and $f_{\text{out}} : \mathbb{R}^d \to \mathbb{R}$ serve as mapping functions for predicting gate-level degrees. This task allows GNNs to infer the connectivity between nodes, providing insights into the interaction patterns for understanding the attributes of each gate and the organization of the circuits. As illustrated in Figure 3, the AIG reconstruction is implemented by gate type prediction and gate-level degree prediction from reconstructed representation for both MGM and VGA. Consequently, the AIG reconstruction loss for MGM and VGA can be defined as:

$$\mathcal{L}_{\text{mgm/vga}} = \mathcal{L}_{\text{type}} + \mathcal{L}_{\text{degree}}. \tag{8}$$

### 3.5 CONSTRAINED MASKED MODELING: MGVGA

Building upon the methodologies of MGM and VGA based on the AIG autoencoder, we propose a novel constrained masked modeling paradigm, MGVGA, to perform general circuit representation learning. This paradigm synthesizes these strategies to develop GNNs that effectively capture diverse and intricate features of circuits. Formally, the loss function for MGVGA can be defined as:

$$\mathcal{L}_{\text{mgvga}} = \mathcal{L}_{\text{mgm}} + \mathcal{L}_{\text{vga}}. \tag{9}$$

This integration enables the GNNs to learn concurrently from fine-grained structural information and abstract circuit function features, optimizing a unified representation that facilitates a wide range of logic synthesis tasks such as classification, regression, and complex reasoning on circuits.

## 4 EXPERIMENTS

### 4.1 DATA PREPARATION

**AIG Collection For MGM.** We obtain 27 circuit designs from five circuit benchmarks as our training dataset: MIT LL Labs CEP (Chetwynd et al., 2019), ITC'99 (Davidson, 1999), IWLS'05 (Albrecht, 2005), EPFL (Amarú et al., 2015), and OpenCore (Takeda, 2008). The resulting AIG dataset comprises 810000 AIGs and 40500 synthesis labels across various optimization sequences and circuit designs. We provide more details of the AIG collection in Appendix A.2.1.

**Verilog-AIG Pairs Collection For VGA.** In this phase, source Verilog codes (Thakur et al., 2023; Liu et al., 2023) are selected and subjected to logic synthesis using Yosys (Wolf et al., 2013), and then they are converted into AIG format. This process yields 64826 Verilog-AIG pairs, which are utilized for VGA illustrated in Section 3.3.

**AIG Preprocessing.** As mentioned previously, we convert circuits to AIGs to implement our MGVGA. The node type of AIG can be categorized into PI, PO, AND, and NOT gates. Given that the number of PIs and POs is typically minimal, the primary emphasis in masked modeling lies in the accurate reconstruction of AND and NOT gates. It is worth noting that the NOT gate is the only single-input logic gate. Our concern is that the model could potentially leverage the disparity in in-degrees of gates as a shortcut, thereby simplifying the reconstruction task without learning the useful circuit representations. Consequently, we introduce single-input AND gates during the training phase as illustrated in Figures 1 to 3, which have two identical inputs. The input and output of the single-input AND gate are identical, making this augmentation simply adaptable to circuits featuring various logic gates (NAND, XOR, OR, etc.). Our experiments indicate that GNNs struggle to precisely capture degree information during the reconstruction process. Consequently, we employ this augmentation method as a trick in our training process, treating it as an equivalent augmentation for AIGs to avoid overfitting and possible leakage. Notably, this augmentation is **not** utilized during the evaluation phase as shown in Figure 5.

**Evaluation Dataset Collection.** As for the evaluation dataset, we select 10 circuit designs external to the training dataset from opensource benchmark (Chowdhury et al., 2021; Amarú et al., 2015; OpenRISC, 2009; YosysHQ, 2020; Asanovic, 2016), the details of which are illustrated in Table 1. Additionally, we will provide more details on benchmark selection in Appendix A.2.4.

## 4.2 IMPLEMENTATION DETAILS

**Training Process of MGVGA.** For the circuit representation learning utilizing our MGVGA paradigm, we utilize DeepGCN (Li et al., 2019; 2020) as the GNN encoder and decoder. As for the LLM, we utilize gte-Qwen2-7B-instruct (Li et al., 2023b), trained with bidirectional attention mechanisms based on Qwen2-7B (Yang et al., 2024), which has a comprehensive understanding of abstract circuit function described in Verilog codes (Liu et al., 2023; Pei et al., 2024; Tsai et al., 2024; Fang et al., 2024). The training process employs a linear learning rate schedule with the Adam optimizer set at a learning rate of $1 \times 10^{-3}$, a weight decay of 0.01, and a batch size of 512. The model is fine-tuned for 3 epochs on 8×A100 GPUs with 80G memory each. Additionally, we provide more details about the model settings in Appendix A.2.3.

**Baseline Selection.** As for the baseline selection, we select DeepGate2 (Shi et al., 2023) as a baseline due to its similar scope and SOTA performance in general circuit representation learning. Additionally, we provide more details of baseline selection in Appendix A.2.2.

**Evaluation.** To validate the efficacy of our MGVGA, we conduct evaluations across two distinct logic synthesis tasks including the Quality of Results (QoR) Prediction and Logic Equivalence Identification tasks. During the evaluation process, we utilize the GNN encoder trained with MGVGA to extract the AIG representation **without** extracting Verilog code representations. Moreover, we extract the AIG representation directly without fine-tuning DeepGate2 and MGVGA for downstream tasks. Notably, QoR prediction aims to assess the ability to extract structural information, whereas logic equivalence identification is designed to evaluate the capability of extracting abstract function information. Moreover, besides identifying logic equivalence directly, we conduct experiments on the boolean satisfiability solving (SAT) task in Appendix A.3.2. SAT solving requires rough logic equivalence checking to be strictly validated later. Additionally, we provide more details about the model settings in Appendix A.2.3.

## 4.3 QoR PREDICTION

For QoR prediction tasks, we estimate the number of optimized gates for the circuit designs following logic synthesis optimization via ABC (Brayton & Mishchenko, 2010). As illustrated in Figure 5(a), we utilize a GNN encoder to extract circuit embeddings exclusively from AIGs. These embeddings are extracted to train the QoR prediction model with the datasets described in Section 4.1. We evaluate the performance across ten circuit designs as illustrated in Table 1, with

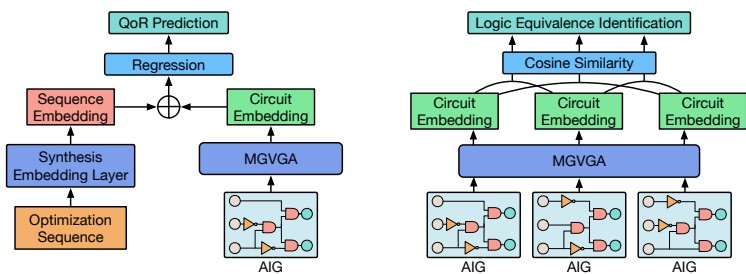

Figure 5: Application of MGVGA in QoR prediction and logic equivalence identification.

Table 1: Performance of DeepGate2 and MGVGA on QoR prediction.

| Design | # PI | # PO | # Gates | DeepGate2 | | | | | MGVGA (Ours) | | | | |
|---|---|---|---|---|---|---|---|---|---|---|---|---|---|
| | | | | NDCG@$k$ ↑ | | Top-$k$% Commonality ↑ | | | NDCG@$k$ ↑ | | Top-$k$% Commonality ↑ | | |
| | | | | $k$=3 | $k$=5 | $k$=3 | $k$=5 | $k$=10 | $k$=3 | $k$=5 | $k$=3 | $k$=5 | $k$=10 |
| bc0 | 21 | 11 | 2784 | 0.331 | 0.395 | 0.244 | 0.227 | 0.280 | 0.444 | 0.560 | 0.222 | 0.213 | 0.320 |
| apex1 | 45 | 45 | 2661 | 0.645 | 0.643 | 0.222 | 0.333 | 0.413 | 0.706 | 0.716 | 0.311 | 0.400 | 0.513 |
| div | 128 | 128 | 101698 | -0.063 | 0.029 | 0.000 | 0.027 | 0.133 | -0.060 | -0.060 | 0.000 | 0.013 | 0.093 |
| k2 | 45 | 45 | 4075 | -0.060 | 0.040 | 0.022 | 0.040 | 0.080 | 0.902 | 0.873 | 0.267 | 0.320 | 0.400 |
| i10 | 257 | 224 | 3618 | -0.133 | -0.080 | 0.000 | 0.000 | 0.027 | 0.620 | 0.607 | 0.289 | 0.307 | 0.353 |
| mainpla | 26 | 49 | 9441 | 0.674 | 0.629 | 0.267 | 0.293 | 0.360 | 0.594 | 0.598 | 0.200 | 0.187 | 0.233 |
| or1200_cpu | 2343 | 2072 | 56570 | 0.498 | 0.485 | 0.178 | 0.267 | 0.407 | 0.617 | 0.613 | 0.222 | 0.253 | 0.367 |
| picorv32 | 1631 | 1601 | 25143 | 0.563 | 0.406 | 0.111 | 0.173 | 0.186 | 0.440 | 0.457 | 0.066 | 0.160 | 0.180 |
| Rocket | 4413 | 4187 | 96507 | 0.578 | 0.543 | 0.111 | 0.186 | 0.300 | 0.557 | 0.607 | 0.355 | 0.413 | 0.467 |
| sqrt | 128 | 64 | 40920 | 0.304 | 0.153 | 0.000 | 0.027 | 0.080 | 0.577 | 0.401 | 0.000 | 0.040 | 0.080 |
| Average | | | | 0.334 | 0.324 | 0.116 | 0.157 | 0.226 | **0.540** | **0.537** | **0.193** | **0.231** | **0.301** |

each circuit undergoing synthesis through 1500 optimization sequences, each containing 20 steps. Notably, we detail the evaluation process and evaluation metrics of QoR prediction task in Appendix A.2.5. Specifically, we utilize two evaluation metrics including NDCG@$k$ for $k = 3, 5$ and Top-$k$% Commonality for $k = 3, 5, 10$ in our experiments.

Table 1 illustrates a detailed comparison between MGVGA and DeepGate2 in QoR prediction, using the post-synthesis number of gates. MGVGA excels in NDCG@$k$ for $k = 3, 5$ and achieves higher percentages in Top-$k$% Commonality for $k = 3, 5, 10$ across various designs. When considering the average performance, MGVGA notably surpasses DeepGate2 with an NDCG@3 of 0.540 compared to 0.334, and a Top-10% Commonality score of 0.301 against 0.226. This demonstrates MGVGA's superior capability in extracting structure information of circuits, which facilitates recommending optimal optimization sequences. Collectively, these results quantitatively validate the significant advancement of MGVGA over DeepGate2 in logic circuit optimization tasks, particularly in accurately and efficiently predicting superior gate configurations to enhance overall design quality.

## 4.4 LOGIC EQUIVALENCE IDENTIFICATION

Each design in Table 2 undergoes optimization to generate various graph expressions while ensuring functionality equivalence. The Cone, specifying the PIs and PO constructs, also denotes functionality equivalence. To evaluate the GNNs' ability to identify circuit function, we derive logically equivalent gates by isolating the Cone among the designs in Table 2 within AIGs using the `cone` command within logic synthesis tool ABC (Brayton & Mishchenko, 2010). Our dataset consists of 10000 pairs of logic gates to test the identification of logic equivalence. As illustrated in Figure 5(b), GNNs deem pairs of logic gates as equivalent if the cosine similarity between their embeddings exceeds a predefined threshold during the evaluation process. The predefined threshold is optimized based on the receiver operating characteristic (ROC) curve. In our experiments, we assess the GNNs' performance using precision, recall, F1-score, and the area under the ROC curve (AUC).

As illustrated in Table 2, our proposed MGVGA has shown significant superiority over the established DeepGate2. The comprehensive analysis reveals that MGVGA outperforms DeepGate2, achieving an F1-score of 0.470 compared to 0.424, and an AUC of 0.862 versus 0.804. Moreover, we also provide more experiment results for the comparison between DeepGate3 and MGVGA using small designs in Appendix A.3.1. These results underscore the consistency of MGVGA and its su-

Table 2: Performance of DeepGate2 and MGVGA on logic equivalence identification.

| Design | DeepGate2 | | | | MGVGA (Ours) | | | |
|---|---|---|---|---|---|---|---|---|
| | Precision | Recall | F1-Score | AUC | Precision | Recall | F1-Score | AUC |
| bc0 | 0.199 | 0.930 | 0.327 | 0.813 | 0.274 | 0.715 | 0.396 | 0.817 |
| apex1 | 0.133 | 0.680 | 0.223 | 0.601 | 0.273 | 0.710 | 0.394 | 0.826 |
| div | 0.203 | 0.980 | 0.337 | 0.814 | 0.197 | 0.670 | 0.305 | 0.725 |
| k2 | 0.171 | 0.720 | 0.276 | 0.695 | 0.336 | 0.920 | 0.492 | 0.919 |
| i10 | 0.414 | 0.940 | 0.575 | 0.918 | 0.699 | 0.950 | 0.805 | 0.985 |
| mainpla | 0.178 | 0.790 | 0.290 | 0.732 | 0.167 | 0.900 | 0.281 | 0.746 |
| or1200_cpu | 0.451 | 0.790 | 0.575 | 0.823 | 0.356 | 0.950 | 0.518 | 0.929 |
| picorv32 | 0.448 | 0.870 | 0.592 | 0.918 | 0.440 | 0.960 | 0.604 | 0.941 |
| Rocket | 0.346 | 0.930 | 0.504 | 0.892 | 0.388 | 1.000 | 0.559 | 0.952 |
| sqrt | 0.189 | 0.740 | 0.302 | 0.721 | 0.199 | 0.720 | 0.312 | 0.770 |
| Average | 0.295 | 0.841 | 0.424 | 0.804 | **0.336** | **0.848** | **0.470** | **0.862** |

Table 3: Performance of MGVGA on QoR prediction and logic equivalence identification with different masking ratios of MGM and VGA.

| Masking Ratio | | QoR Prediction | | | | | Logic Equivalence Identification | | | |
|---|---|---|---|---|---|---|---|---|---|---|
| MGM | VGA | NDCG@$k$ ↑ | | Top-$k$% Commonality ↑ | | | Precision | Recall | F1-Score | AUC |
| | | $k=3$ | $k=5$ | $k=3$ | $k=5$ | $k=10$ | | | | |
| 0.3 | 0.3 | 0.517 | 0.505 | 0.158 | 0.199 | 0.272 | 0.300 | 0.844 | 0.430 | 0.846 |
| 0.3 | 0.5 | **0.540** | **0.537** | **0.193** | **0.231** | **0.301** | **0.336** | 0.848 | **0.470** | **0.862** |
| 0.3 | 0.7 | 0.498 | 0.514 | 0.178 | 0.223 | 0.299 | 0.316 | 0.823 | 0.441 | 0.820 |
| 0.5 | 0.3 | 0.439 | 0.445 | 0.149 | 0.187 | 0.271 | 0.274 | 0.821 | 0.402 | 0.821 |
| 0.5 | 0.5 | 0.438 | 0.470 | 0.169 | 0.204 | 0.288 | 0.331 | 0.829 | 0.450 | 0.836 |
| 0.5 | 0.7 | 0.385 | 0.415 | 0.140 | 0.168 | 0.247 | 0.304 | 0.833 | 0.433 | 0.817 |
| 0.7 | 0.3 | 0.347 | 0.367 | 0.127 | 0.160 | 0.242 | 0.292 | **0.871** | 0.421 | 0.828 |
| 0.7 | 0.5 | 0.400 | 0.366 | 0.111 | 0.175 | 0.228 | 0.313 | 0.823 | 0.433 | 0.832 |
| 0.7 | 0.7 | 0.366 | 0.359 | 0.138 | 0.169 | 0.226 | 0.305 | 0.794 | 0.425 | 0.822 |

perior ability to accurately determine functionally equivalent circuits, highlighting its effectiveness in extracting abstract circuit function features.

## 4.5 ANALYSIS OF MASKING RATIO

This section analyzes the impact of masking ratios for both MGM and VGA. Table 3 indicates that the optimal masking ratio for MGM is 0.3, while the optimal masking ratio for VGA is 0.5. At an MGM masking ratio of 0.3, the MGVGA method demonstrates notable performance. Meanwhile, there is a consistent improvement in both QoR prediction and logic equivalence identification tasks when the VGA masking ratio increases from 0.3 to 0.5.

The study reveals that higher MGM masking ratios negatively affect QoR performance, suggesting that excessive masking impedes the GNNs' ability to learn information effectively. Specifically, excessively high masking ratios (0.5 and 0.7) significantly reduce the performance of MGVGA in the QoR prediction task, which reflects the capability in structural circuit information extraction. As for VGA, a relatively higher masking ratio (0.5) generally yields better performance for the logic equivalence identification task. An excessively high masking ratio (0.7) degrades the performance of extracting circuit structural and functional features. These findings align with our intuitive expectations. MGM directly enables GNNs to recover complete structural information from masked latent space. GNNs can not learn effective information from unmasked gates if the masking ratio is too high. In the VGA task, introducing Verilog codes as constraints for circuit restoration allows for a relatively high masking ratio, as Verilog codes contain rich circuit information.

## 4.6 EFFECTIVENESS OF CONSTRAINT MASKED MODELING

To assess the effectiveness of our MGM and VGA approaches, we conduct an ablation study on QoR prediction and logic equivalence identification tasks. We use the original masked modeling strategy (Hou et al., 2023) as a baseline, employing a masking ratio of 0.3 for both the original and MGM methods, based on the optimal performance observed in Table 3.

As shown in Table 4, the original masked modeling strategy yielded poor performance in both tasks. This outcome aligns with our previous assertion that masking in the original circuits disrupts their logical equivalence, thereby preventing the method from learning effective circuit features. In con-

Table 4: Ablation study on constraint masked modeling paradigm, including MGM and VGA.

| Mask Strategy | | QoR Prediction | | | | | Logic Equivalence Identification | | | |
|---|---|---|---|---|---|---|---|---|---|---|
| MGM | VGA | NDCG@$k$ ↑ | | Top-$k$% Commonality ↑ | | | Precision | Recall | F1-Score | AUC |
| | | $k$=3 | $k$=5 | $k$=3 | $k$=5 | $k$=10 | | | | |
| ✗ | ✗ | 0.153 | 0.207 | 0.107 | 0.159 | 0.255 | 0.264 | 0.793 | 0.382 | 0.790 |
| ✔ | ✗ | 0.338 | 0.368 | 0.135 | 0.197 | 0.289 | 0.305 | **0.850** | 0.433 | 0.833 |
| ✔ | ✔ | **0.540** | **0.537** | **0.193** | **0.231** | **0.301** | **0.336** | 0.848 | **0.470** | **0.862** |

Table 5: Generalization on various GNNs, including GraphSAGE and graph transformer.

| GNNs | QoR Prediction | | | | | Logic Equivalence Identification | | | |
|---|---|---|---|---|---|---|---|---|---|
| | NDCG@$k$ ↑ | | Top-$k$% Commonality ↑ | | | Precision | Recall | F1-Score | AUC |
| | $k$=3 | $k$=5 | $k$=3 | $k$=5 | $k$=10 | | | | |
| DeepGate2 | 0.334 | 0.324 | 0.116 | 0.157 | 0.226 | 0.295 | 0.841 | 0.424 | 0.804 |
| GraphSAGE | 0.469 | 0.479 | 0.153 | 0.224 | **0.314** | 0.329 | 0.811 | 0.455 | 0.841 |
| Graph Transformer | 0.452 | 0.470 | 0.154 | 0.212 | 0.311 | 0.324 | 0.789 | 0.450 | 0.831 |
| DeepGCN (Ours) | **0.540** | **0.537** | **0.193** | **0.231** | 0.301 | **0.336** | **0.848** | **0.470** | **0.862** |

trast, both our MGM and VGA approaches demonstrated significant improvements in the two tasks, underscoring their effectiveness in enhancing GNNs' capacity to extract fine-grained structural information and abstract functional data. Furthermore, VGA not only facilitates GNNs' extraction of circuit function features but also enhances their ability to recognize circuit structure during the reconstruction process. This improvement occurs under the constraint of logically equivalent Verilog codes, highlighting the versatility of our VGA approach.

### 4.7 GENERALIZATION ON VARIOUS GNNS

To evaluate the generalization capability of our MGVGA, we perform circuit representation learning using various traditional GNNs, including GraphSAGE (Hamilton et al., 2017) and graph transformer (Shi et al., 2021). Similarly, based on the optimal performance observed in Table 3, where MGVGA achieves the best results with the MGM masking ratio of 0.3 and the VGA masking ratio of 0.5, we apply these same masking ratios to the constrained masked modeling of other GNNs.

As shown in Table 5, all GNNs trained with MGVGA exhibited significant improvements compared to the baseline DeepGate2 model. These results demonstrate the exceptional generalization ability of our proposed methods across different GNN architectures. This consistent performance enhancement across various models underscores the robustness and versatility of our MGVGA paradigm in extracting better circuit representation via circuit representation learning.

## 5 CONCLUSION

In conclusion, this study introduces a novel constrained mask modeling paradigm, MGVGA, for circuit representation learning. This method integrates MGM and VGA to extract fine-grained structural information and abstract functions of circuits. MGM operates by masking gates in the latent space rather than in the original circuits, subsequently reconstructing the attributes of these masked gates. This approach preserves the logical equivalence of circuits, overcoming the limitations of traditional masked gate modeling strategies. Moreover, we developed VGA, which performs masking operations on the original circuits and reconstructs them under the constraints of equivalent Verilog codes. This enables GNNs to learn circuit functions from LLMs. Our comprehensive evaluations demonstrate that MGVGA performs better than previous SOTA methods in QoR prediction and logic equivalent identification tasks. This represents a significant advancement in applying the constrained masked modeling paradigm to general circuit representation learning.

### ACKNOWLEDGEMENTS

This work is partially supported by The Research Grants Council of Hong Kong SAR (No. RFS2425-4S02, No. CUHK14211824, No. CUHK14210723), AI Chip Center for Emerging Smart Systems (ACCESS), Hong Kong SAR, and Shanghai Artificial Intelligence Laboratory.

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

# A    APPENDIX

## A.1    MORE DETAILS OF MGVGA

### A.1.1    LOGIC EQUIVALENCE PRESERVATION

In traditional masked graph modeling processes, nodes in graphs are masked directly and then reconstructed without any constraint except labels of masked nodes. However, any valid circuits can be labeled during the reconstruction process for circuit representation learning. It's hard for GNNs to learn useful features for downstream tasks in this way. Consequently, we introduce constraints during the decoding process to ensure that GNNs learn useful features related to the circuit.

In our work, "logical equivalence preservation" describes the equivalence between an AIG $\mathcal{G} = (\mathcal{V}, \mathcal{A})$ and its different representations including $X = g_E(\mathcal{V}, \mathcal{A})$ and $X_V$. In **MGM**, $\mathcal{G}$ and its latent space embedding $X$ are logically equivalent. According to $X = g_E(\mathcal{V}, \mathcal{A})$, the latent space embedding $X$ is derived from $\mathcal{G}$ through the GNN encoding process without masking gates(nodes). Consequently, we can say that $X$ and $\mathcal{G}$ comes from the same truth table. In **VGA**, $\mathcal{G}$ and its corresponding Verilog code embedding $X_V$ are logically equivalent. $X_V$ is extracted via LLM according to the given Verilog code and $\mathcal{G}$ is obtained from the Verilog code via logic synthesis tools as illustrated in Figure 2. Similarly, we can say that the truth tables of $X_V$ and $\mathcal{G}$ are the same.

In summary, we won't change the original structure of AIG $\mathcal{G}$ during the reconstruction process and the logic information is retained in $X$ or $X_V$ as the constraint for the decoding process.

### A.1.2    UNMASKED ADJACENCY MATRIX

As we mentioned, following previous masked modeling paradigms (Hou et al., 2022; 2023), we do **not** mask the adjacent matrices $\mathcal{A}$ during the entire process. Here are the detailed reasons.

**Logical Equivalence:** Our MGVGA emphasizes logical equivalence during training, ensuring that the circuit has a unique solution during reconstruction. If both $\mathcal{A}$ and $X$ were masked, the problem would become NP-hard due to multiple possible solutions, making it much more difficult to train the model effectively and convergently.

**Computational Complexity:** Reconstructing the adjacency matrix $\mathcal{A}$ would require handling a matrix of size $N^2$ for $N$ gates, which is computationally infeasible for large circuits (e.g., those with millions of gates). By not masking $\mathcal{A}$, we only need to reconstruct the masked gate information in $X$, significantly reducing computational complexity and improving the efficiency of both training and inference. Moreover, both MGM and VGA in MGVGA operate at the gate level without masking the adjacency matrix $\mathcal{A}$, focusing on local feature extraction. Consequently, we can utilize parallel processing techniques, distributed computing, etc., to reduce overhead in both MGM and VGA stages for high-complexity circuits.

In summary, we perform the MGVGA without masking the adjacency matrix $\mathcal{A}$, which is the same as previous masked graph modeling methods (Hou et al., 2022; 2023). Consequently, there are only two tasks for gate-level prediction to reconstruct masked circuits as illustrated in Section 3.4.

## A.2    MORE DETAILS OF EXPERIMENT SETTINGS

### A.2.1    AIG COLLECTION

Yosys (Wolf et al., 2013) is utilized to conduct logic synthesis, which converts source codes of circuit designs into the standardized AIG format. Moreover, we prepare 1500 optimization sequences, each containing 20 synthesis transformations including `rewrite`, `resub`, `refactor`, `rewrite -z`, `resub -z`, `refactor -z`, and `balance` transformations, consistent with prior works (Chowdhury et al., 2022; Zheng et al., 2024). Then sequential synthesis transformations are carried out by the logic synthesis tool ABC (Brayton & Mishchenko, 2010) and their corresponding labels are generated. Meanwhile, we also store the AIG after each synthesis transformation for AIG self-supervised learning. The resulting AIG dataset, post-technology mapping with the Nan-Gate 45nm technology library and the "5K heavy 1k" wireload model, comprises 810000 AIGs and 40500 synthesis labels across various optimization sequences and circuit designs.

### A.2.2 BASELINE SELECTION

During the baseline selection process, we acknowledged and recognized the progress made with DeepGate3 (Shi et al., 2024), the upgraded version of DeepGate2 (Shi et al., 2023). However, we encountered several challenges in our practical implementation.

Specifically, DeepGate3's architecture is built upon DeepGate2 and incorporates transformer models, which have a quadratic time complexity during attention computation. This becomes a critical issue when dealing with large-scale datasets. Our training set includes digital circuits with up to millions of gates, leading to substantial and often unmanageable computational costs during training. During the testing phase, DeepGate3 is limited to circuits with up to thousands of gates. When attempting to infer circuits with millions of gates, the computational overhead becomes prohibitively high. This limitation makes it impractical for our use case, where we need to handle circuits of varying sizes, including very large ones.

Consequently, we choose DeepGate2 as our baseline because it's the best model that provides a more practical and scalable solution for the digital circuits we aim to model and optimize within the EDA toolchain. Moreover, we also provide some experiment results for the comparison between DeepGate3 and MGVGA on the small-scale designs with just thousands of gates in Appendix A.3.

### A.2.3 MORE DETAILS OF MODEL SETTINGS

**Model Size.** DeepGate2 (Shi et al., 2023) has 0.64M parameters with 1 layer and DeepGate3 (Shi et al., 2024) has 8.17M parameters with transformer architecture. Meanwhile, our MGVGA has only 0.12M parameters with 7 layers. According to the experiment results, MGVGA achieved much better performance compared to DeepGate2 and DeepGate3 with fewer model parameters, which demonstrates the effectiveness of our method.

**Bidirectional LLM.** We utilize gte-Qwen2-7B-instruct (Li et al., 2023b) model for Verilog code representation extraction, which is based on a BERT-like encoder transformer architecture with bidirectional attention. The base model of gte-Qwen2-7B-instruct, Qwen2-7B-instruct (Yang et al., 2024), is a decoder-based model with causal attention. Qwen2-7B-instruct has been extensively trained on a diverse corpus of Verilog and demonstrates an extraordinary ability to understand and process various styles and complexities of Verilog/System Verilog code, including less standardized or non-optimized representations. Although it excels in generating text and understanding sequential dependencies, it is not well-suited for embedding tasks due to its unidirectional attention mechanism. Consequently, Li et al. (2023b) proposes GTE to transform Qwen2-7B-instruct into gte-Qwen2-7B-instruct, enabling the model to capture bidirectional context while preserving the original capabilities in understanding Verilog codes.

### A.2.4 BENCHMARK SELECTION

Given the practical requirements of our application, we chose DeepGate2 as our baseline. DeepGate2 uses a portion of the design from open-source benchmarks as training data. To ensure fair and reliable testing, we excluded these designs from our test set. To further validate the reliability and practicality of our method, we have supplemented our test set with designs from different opensource benchmarks (Chowdhury et al., 2021; Amarú et al., 2015; OpenRISC, 2009; YosysHQ, 2020; Asanovic, 2016) that were not used during training. These additional test sets cover a range of graph sizes and complexities, ensuring that our evaluation is comprehensive and representative of real-world EDA tool requirements.

### A.2.5 EVALUATION DETAILS OF QoR PREDICTION

For the evaluation of the QoR prediction task, we evaluate the performance across ten circuit designs as illustrated in Table 1, with each circuit undergoing synthesis through 1500 optimization sequences, each containing 20 steps. Notably, normalization is applied to $\mathbf{A} \in \mathbb{R}^{1 \times 1500}$, representing the count of optimized gates per sequence, following (Chowdhury et al., 2021). Specifically, each element $\mathbf{A}_i$ is standardized using $\mathbf{A}_i = \frac{\bar{\mathbf{A}} - \mathbf{A}_i}{\sigma_{\mathbf{A}}}$, where $\bar{\mathbf{A}}$ and $\sigma_{\mathbf{A}}$ is the mean and the standard deviation of $\mathbf{A}$, respectively. For QoR prediction, we need to rank the predicted scores $\mathbf{B} \in \mathbb{R}^{1 \times 1500}$ to identify the best optimization sequence. Consequently, we utilize the Normalized Discounted

Table 6: Performance of DeepGate models and MGVGA on logic equivalence identification.

| Design | # PI | # PO | # Gates | DeepGate2 | | DeepGate3 | | MGVGA (Ours) | |
|---|---|---|---|---|---|---|---|---|---|
| | | | | F1-Score | AUC | F1-Score | AUC | F1-Score | AUC |
| bc0 | 21 | 11 | 2784 | 0.327 | 0.813 | 0.373 | 0.819 | 0.396 | 0.817 |
| apex1 | 45 | 45 | 2661 | 0.223 | 0.601 | 0.326 | 0.725 | 0.394 | 0.826 |
| k2 | 45 | 45 | 4075 | 0.276 | 0.695 | 0.345 | 0.834 | 0.492 | 0.919 |
| i10 | 257 | 224 | 3618 | 0.575 | 0.918 | 0.601 | 0.928 | 0.805 | 0.985 |
| mainpla | 26 | 49 | 9441 | 0.290 | 0.732 | 0.305 | 0.763 | 0.281 | 0.746 |
| Average | | | | 0.338 | 0.752 | 0.390 | 0.834 | **0.474** | **0.859** |

Cumulative Gain (NDCG) (Järvelin & Kekäläinen, 2017; Järvelin & Kekäläinen, 2002) metric to assess the quality of the ranking algorithms for the predicted scores $\mathbf{B}$. The NDCG@$k$ is calculated as follows:

$$\text{NDCG@}k = (\sum_{i=1}^{k} \frac{\mathbf{A}_{\text{rank}(\mathbf{B},i)}}{\log_2(i+1)}) / (\sum_{i=1}^{k} \frac{\mathbf{A}_{\text{rank}(\mathbf{A},i)}}{\log_2(i+1)}), \tag{10}$$

where $k$ represents the position considered in the ranking. Here, $\text{rank}(\mathbf{A},i)$ and $\text{rank}(\mathbf{B},i)$ denote the indices in $\mathbf{A}$ and $\mathbf{B}$ of the $i$-th largest elements, respectively. The NDCG@$k$ score ranges from -1 to 1, with a higher score indicating better ranking performance. A perfect ranking would achieve an NDCG@$k$ score of 1. Furthermore, we evaluate and compare the predictions on a reference set of optimization sequences with actual synthesis labels using the Top-$k\%$ Commonality metrics, defined as $\frac{\text{num}(\tilde{\mathbf{A}}_k \cap \tilde{\mathbf{B}}_k)}{\text{num}(\tilde{\mathbf{A}}_k)}$, where $\tilde{\mathbf{A}}_k$ and $\tilde{\mathbf{B}}_k$ represent the top $k\%$ performing optimization sequences, actual and predicted, respectively.

## A.3 EXTENDED EXPERIMENTAL RESULTS

### A.3.1 LOGIC EQUIVALENCE IDENTIFICATION

As shown in Table 6, we present the performance comparison between DeepGate models (Shi et al., 2023; 2024) and MGVGA on small circuit designs considering the computation overhead. The experiment results show that MGVGA outperforms DeepGate3, achieving an average F1-score of 0.474 compared to 0.390, and an average AUC of 0.859 versus 0.834.

### A.3.2 BOOLEAN SATISFIABILITY SOLVING

Boolean satisfiability (SAT) solving aims to determine whether there exists an assignment of truth values that satisfies a Boolean formula. In logic synthesis, SAT solvers are indispensable for tasks such as logic optimization, ensuring both the correctness and efficiency of circuits. Despite their critical role, SAT solving remains computationally challenging, often leading to significant runtime overhead, especially for large or complex designs. To address this challenge, several studies (Haaswijk et al., 2019; Shi et al., 2023; 2024) have proposed various methods to accelerate the SAT-solving process. For instance, Exact synthesis (Haaswijk et al., 2019) enhanced SAT solving by systematically varying the number of nodes and levels in directed acyclic graph (DAG) topologies, a technique referred to as Boolean fences. A Boolean fence is a partition of nodes across multiple levels, with each level containing at least one node. By tuning these parameters, they effectively constrained the search space for the SAT solver, resulting in more efficient and predictable synthesis outcomes. Despite achieving significant speedup over SAT solvers, the solution still has room for further enhancement. Building on the work presented in (Haaswijk et al., 2019), we demonstrate how MGMVA can efficiently accelerate the SAT-solving process.

**Exact Synthesis** generates logic circuits that guarantee logical equivalence between the resulting circuit and the target logic function, intending to find an optimal implementation based on specific criteria, such as gate count or depth.

**Experiment Settings.** We integrate the MGMVA into the SAT solver (Haaswijk et al., 2019) to perform exact synthesis tasks. For the training process, we apply the exact synthesis process to the EPFL (Amarú et al., 2015) and IWLS (Albrecht, 2005) benchmarks using the SAT solver (Haaswijk et al., 2019) to obtain the number of nodes and levels in the subcircuits, which are subsequently used as labels. We extract the circuit embeddings from MGVGA and feed them into MLP to perform regression tasks. As for the SAT-solving process with MGVGA, we first extract several small

Table 7: The comparison of SAT solving runtime (solver only).

| Design | # Subcircuits | Exact Synthesis | DeepGate2 | | DeepGate3 | | MGVGA (Ours) | |
|--------|---------------|-----------------|-----------|------|-----------|------|--------------|------|
| | | | Solver | Red. | Solver | Red. | Solver | Red. |
| adder | 27 | 365.19 | 480.72 | -31.64% | 464.21 | -27.11% | 184.19 | 49.56% |
| sqrt | 38 | 5.55 | 6.65 | -19.82% | 6.92 | -24.68% | 4.77 | 14.05% |
| hyp | 80 | 328.58 | 351.25 | -6.90% | 351.71 | -7.04% | 213.20 | 35.11% |
| i2c | 169 | 267.15 | 62.21 | 76.71% | 34.01 | 87.27% | 32.98 | 87.65% |
| div | 1968 | 4033.48 | 1096.21 | 72.83& | 882.59 | 78.12% | 844.45 | 79.06% |
| Average | | | 399.41 | 18.24% | 347.89 | 21.31% | **255.92** | **53.09%** |

Table 8: The comparison of SAT solving runtime (overall).

| Design | Exact Synthesis | DeepGate2 | | | DeepGate3 | | | MGVGA (Ours) | | |
|--------|-----------------|-------|--------|---------|-------|--------|---------|-------|--------|---------|
| | | Model | Solver | Overall | Model | Solver | Overall | Model | Solver | Overall |
| adder | 365.19 | 0.66 | 480.72 | 481.38 | 10.86 | 464.21 | 475.07 | 0.17 | 184.19 | 184.36 |
| sqrt | 5.55 | 0.65 | 6.65 | 7.30 | 10.91 | 6.92 | 17.83 | 0.24 | 4.77 | 5.01 |
| hyp | 328.58 | 2.13 | 351.25 | 353.38 | 36.01 | 351.71 | 387.72 | 1.32 | 213.20 | 215.84 |
| i2c | 267.15 | 2.60 | 62.21 | 64.81 | 45.95 | 34.01 | 79.96 | 1.06 | 32.98 | 34.04 |
| div | 4033.48 | 40.80 | 1096.21 | 1137.01 | 763.71 | 882.89 | 1646.60 | 17.11 | 844.45 | 861.56 |
| Average | 999.99 | 9.37 | 399.41 | 408.78 | 173.49 | 347.89 | 521.38 | **3.99** | **255.92** | **259.91** |

subcircuits from the original circuits following the exact synthesis process. These subcircuits, which have a maximum of 8 inputs and a single output, are still computationally challenging due to the exponential number of Boolean functions $2^{256}$ resulting in large runtime overhead. Then, we can use the MGMVA to predict the number of nodes and levels required for the optimal equivalent implementation of the input subcircuit and use MGMVA to further constrain the search space of the Boolean fence, thereby accelerating the SAT-solving process. Finally, to assess the efficacy of our model in accelerating SAT solving, we employ DeepGate2 (Shi et al., 2023) and DeepGate3 (Shi et al., 2024) as baselines. All experiments are conducted using the same computational resources.

**Evaluation Results.** Table 7 and Table 8 present a runtime comparison among the exact synthesis, DeepGate2, DeepGate3, and our MGVGA settings, with runtime measured in **seconds (s)**. We choose exact synthesis as our baseline setting the runtime reduction compared to the baseline setting is denoted as Red. As shown in Table 7, MGVGA achieves an average runtime reduction of 53.09% while DeepGate2 and DeepGate3 achieve an average reduction of 18.24% and 21.31% separately for SAT solving process. However, it is worth noting that the SAT solver with GNN is less effective for easier cases, as the model inference process accounts for a significant portion of the total runtime as illustrated in Table 8. In general, MGVGA exhibits significant improvement compared to DeepGate models in this task, indicating that MGVGA can capture more informative abstract functional and fine-grained structural representations for solving practical SAT problems.

