# OpenReview forum: "Circuit Representation Learning with Masked Gate Modeling and Verilog-AIG Alignment"
_ICLR.cc/2025/Conference — ICLR 2025 Poster_

### Official Review · Reviewer_EYP6 · 2024-10-28

**Soundness:** 2
**Presentation:** 2
**Contribution:** 3
**Rating:** 6
**Confidence:** 3

**Summary:**

This paper consists of two methods to auto-encode an AIG circuit. The first approach (MGM) encodes an AIG with masking. Rather than masking AIG nodes, it first transform the AIG to $N$ vectors in which $N$ is the number of nodes, and then replace some of the vectors to a "mask vector" $m$. The second approach leverages the Verilog code of the AIG. It first masks some AIG nodes and transforms the masked AIG to another $N$ vectors, and then encode the corresponding Verilog code by LLM. The $N$ vectors and the encoded Verilog code are then merged by a neural block to obtain the AIG representation. Experimental results show that the proposed methods outperformed the baseline method (DeepGate2) on two specific tasks.

**Strengths:**

1. As far as I know, this is the first work to integrate an LLM into circuit representation learning, and if I understand correctly, the method only requires the LLM to engage during the training stage.
2. Improvement over the baseline method on two specific tasks (QoR prediction & logic equivalence identification)
3. Some ablation studies are provided.

**Weaknesses:**

Some key information is omitted, not highlighted enough, or not presented in a clear way, which impede a thorough understanding of this paper, including

1. The preservation of logical equivalence is highlighted multiple times (line 74, 90, 101, 531), but with no detailed explanation/proof/example on how this is achieved.
2. How the AIG is reconstructed from latent space representation $\mathbf{X}$? More specifically, if the encoder is $\mathbf{X} = g_E(\mathcal{V}, \mathcal{A})$, why the decoder is not something like $(\mathcal{V}', \mathcal{A}') = g_D(\mathbf{X})$, but formulated as Equation (2) instead? The adjecency matrix $\mathcal{A}$, which I believe should be the output of a typical reconstruction process, serves as the input of Equation (2) without justification, which confuse me a lot. I also checked Section 3.4 but there are only two tasks on the reconstructed circuit, without explanation of how the reconstructed circuit comes out.
3. In line 252, $X_V \in \mathbb{R}^{1\times d_v}\$ and it works as both key and value, does that mean there is only one key and one value in the cross attention block shown in Figure 4? It is pretty weird to have only a single key-value pair in the cross attention block, since the output will trivially be the value itself.
3. Figure 3 shows the training process of MGVGA, however, it is not clear how MGVGA module (as shown in Figure 5) works in the inference stage. More specifically, Figure 3 includes equivalent Verilog codes for the input outputs, which I hypothesis should not be required during inference, but I cannot confirm it through the paper.
4. In Section 4.2, while the training parameters are provided, the paper does not contain enough details of the model itself. For example, how many layers are included and what is the total number of parameters? Such details are also missing for the baseline method (DeepGate2). Generally speaking, I would like to confirm whether the size of all the compared models are close to each other. For example, in the reserach of LLM, a 7B model is usually compared with other 7B models rather than a 405B model.

**Questions:**

Questions:
1. In Figure 1, why some of the AND nodes have only a single input?
2. In line 71, "logical correctness can still be maintained without necessarily preserving logical equivalence", what does "logical correctness" mean? It seems different from "logical equivalence", which might mean that the circuit is valid (like, no cycles, no NOT node has multiple inputs, etc.)

Some other questions can be found in the "Weaknesses" section above.

Suggesstions on additional contents that may help clarify how the proposed method works:
1. Some detailed explanations/proofs/examples on how the preservation of logical equivalence is achieved.
2. Details on the reconstruction process of the AIG (the process from $\mathbf{X}$ to $(\mathcal{V'}, \mathcal{A'})$, for which the labels in Section 3.4 like $\mathbf{D}^-$ and $\mathbf{D}^+$ can be computed and backpropagated)
3. How MGVGA works in inference. If the equivalent Verilog code in Figure 3 is not required, highlight it.
4. More details in the experimental section about the proposed and compared model (especially the model size and number of layers)

Some other suggestions:

5. The mathematical notations may be marked on Figure 4 for clearness.
6. Instead of identifying logic equivalence directly, it might be of more practical meaning to conduct experiments on downstream tasks requiring "rough" logic equivalence checking to be strictly validated later, such as SAT solving.

---

> ### Comment · Reviewer_EYP6 · 2024-11-23
>
> As some key information is further clarified and revised, I have raised my score.
>
> However, I would still encourage the authors to further improve the presentation and/or release their code. At the current state, I feel like it will still be challenging to reproduce the result.

---

> ### Author Response · Authors · 2024-11-24
>
> Thanks very much for **Reviewer EYP6**'s comments. We are glad that we were able to clarify and revise some key information as requested. We are very grateful for the detailed review comments and suggestions on our paper, which have greatly improved the quality of our work. we deeply appreciate the time and effort the **Reviewer EYP6** has invested in the review process, as well as the highly responsible approach to reviewing. We promise to open-source the entire code of MGVGA including the data preprocessing and model training codes, the pre-trained model weights, and the training logs for reproduction of our methods after publication. Please let us know if there are any other concerns or questions during the discussion period, and we will be happy to clarify.
>
> Once again, thanks very much for the **Reviewer EYP6**'s insightful comments and valuable feedback sincerely.

---

### Official Review · Reviewer_jhYv · 2024-11-03

**Soundness:** 3
**Presentation:** 3
**Contribution:** 2
**Rating:** 5
**Confidence:** 4

**Summary:**

This paper introduces MGVGA, a constrained masked modeling paradigm incorporating masked gate modeling and Verilog-AIG alignment for circuit representation learning. Specifically, MGVGA reserves logical equivalence by masking gates in the latent space rather than in the original circuits, and reconstructs masked gates under the constraints of equivalent Verilog codes. Experiments demonstrate the effectiveness of MGVGA.

**Strengths:**

1.	This paper focuses on circuit representation learning, which is an important problem when incorporating machine learning into many downstream tasks, such as logic synthesis and physical design.
2.	The idea of using the embedding of LLMs to teach GNNs seems reasonable.
3.	Experiments demonstrate the effectiveness of MGVGA.

**Weaknesses:**

1.	The motivation of leveraging the masked modeling paradigm for learning circuit representations is unclear.
2.	The novelty of the proposed method seems incremental. The authors first apply the masked graph autoencoder to learning circuit representations, and align the learned embedding with LLMs. However, the masked graph autoencoder is a relatively mature technique, and aligning embeddings with LLMs has been widely investigated in previous work [1,2,3,4].
3.	Experiments are insufficient. First, some important baselines are missing. It would be more convincing if the authors could compare their method with the recent SOTA method HOGA [5], and the typical GNNs, such as GCN, GAT, etc. Second, the authors may want to conduct experiments on the OpenABC-D benchmark, which is widely used in previous work [5].

[1] Wang, Duo, et al. "LLMs as Zero-shot Graph Learners: Alignment of GNN Representations with LLM Token Embeddings." arXiv preprint arXiv:2408.14512 (2024).

[2] Li, Yuhan, et al. "A survey of graph meets large language model: Progress and future directions." arXiv preprint arXiv:2311.12399 (2023).

[3] Radford, Alec, et al. "Learning transferable visual models from natural language supervision." International conference on machine learning. PMLR, 2021.

[4] Liu, Shengchao, et al. "Multi-modal molecule structure–text model for text-based retrieval and editing." Nature Machine Intelligence 5.12 (2023): 1447-1457.

[5] Deng, Chenhui, et al. "Less is More: Hop-Wise Graph Attention for Scalable and Generalizable Learning on Circuits." DAC 2024.

**Questions:**

Please refer to weaknesses for details.

---

### Official Review · Reviewer_rsFP · 2024-11-04

**Soundness:** 4
**Presentation:** 4
**Contribution:** 4
**Rating:** 8
**Confidence:** 3

**Summary:**

The MGVGA paper explores how to augment Masked Graph Modelling of circuits with Verilog-AIG alignment so that the resulting trained GNN-encoder will be able to encode both graph level circuit-topology information as well as higher level information from the Verilog representation.

The paper then shows its performance against DeepGate2 on various tasks -- such as logic-equivalence and QoR -- to show how MGVGA improves upon the SOTA in this field.

**Strengths:**

This paper very clearly introduces the challenges of applying direct masked-gate-modeling for circuits as well as their decision to perform latent space masking and combining the loss function with a Verilog representation of the circuit for adding more higher-level/abstract information to the output of the encoding.

The paper evaluates against existing SOTA on logic equivalence, one of DeepGate2's tasks as well, thus a solid benchmark.

**Weaknesses:**

Figures 3 may benefit from additional legends and annotation perhaps indicating how the model is concurrently trained (details of the loss function).

If Figure 4 was closer to Figure 3 (perhaps below Figure 3) it would help with seeing precisely when the Verilog comes into the picture.

A figure illustrating Qwen-2 and how it is augmented (or plugged-in) to the training process might be needed to clarify for the reader how this is wired in as a teacher, generally the Verilog-AIG section may benefit from additional diagrams depicting how the LLM integrates with the larger training setup.

Also the "bidirectional attention mechanism" may benefit from a figure or a few lines further to discuss this feature of the gte-Qwen2-7B-instruct model chosen for the Verilog-AIG.

**Questions:**

"As for the LLM, we utilize gte-Qwen2-7B-instruct (Li et al., 2023b), trained with bidirectional attention
mechanisms based on Qwen2-7B (Yang et al., 2024), which has a comprehensive understanding of
abstract circuit function described in Verilog codes (Liu et al., 2023b; Pei et al., 2024)."

What is the "bidirectional attention mechanism" from gte-Qwen2-7B-instruct. While I understand the utility of bidirectional attention in this context, curious to the details on the implementation -- would this perhaps be a BERT-like Encoder Transformer or perhaps a Encoder-Decoder Style approach?

---

### Official Review · Reviewer_xqkr · 2024-11-04

**Soundness:** 3
**Presentation:** 2
**Contribution:** 3
**Rating:** 6
**Confidence:** 3

**Summary:**

The paper introduce MGVGA, a constrained masked modeling framework for circuit representation learning that combines Masked Gate Modeling (MGM) and Verilog-AIG Alignment (VGA). MGM masks gates in the latent space instead of the original circuit to preserve logical equivalence. VGA uses Verilog code as a constraint, allowing graph neural networks (GNNs) to capture circuit functions alongside structural information. The framework aims to leverage the strengths of large language models (LLMs) for understanding circuit functionality from Verilog code and addresses traditional challenges in masked modeling for logic synthesis tasks. Experimental results indicate MGVGA's superior performance in Quality of Results (QoR) prediction and logic equivalence identification compared to state-of-the-art.

**Strengths:**

1. The framework introduces a novel way to apply masked modeling to circuits without compromising logical equivalence, which has been a critical limitation of prior masked modeling approaches. The combination of MGM for structure and VGA for function is effective for circuit representation.

2. The paper uses Verilog-AIG alignment, that  enables GNNs to learn abstract circuit functionalities beyond structural layouts and leverages the representational power of LLMs for Verilog. This step improves the model’s ability to handle complex circuit functions.

3. MGVGA demonstrates significant improvements over DeepGate2 in QoR prediction and logic equivalence identification, validating its efficacy for circuit representation tasks.

**Weaknesses:**

1. although the framework performs well for circuits of moderate complexity, it is unclear how well it scales to larger circuit representations, especially in terms of maintaining efficient reconstruction performance. The masking techniques could face computational bottlenecks when applied to large-scale circuit datasets.
2. Deepgate3[1] is already release and has significantly better performance than deepgate2. The author should compare to the latest results.

[1] DeepGate3: Towards Scalable Circuit Representation Learning

**Questions:**

1. How does MGVGA handle variations in Verilog/system verilog code style and complexity, and is there a performance threshold for handling less standardized or non-optimized Verilog representations?

2. What are the computational implications of applying MGVGA to larger circuits, and are there strategies for reducing overhead in both MGM and VGA stages for high-complexity circuits?

3. Could the MGVGA framework be adapted to support other types of hardware description languages (HDLs) beyond Verilog, and if so, what modifications would be necessary?

---

### Meta-Review · Area_Chair_Lciu · 2024-12-19

**Metareview:**

The paper introduces MGVGA, a novel and constrained masked modeling framework for circuit representation learning that demonstrates superior performance on tasks like QoR prediction and logic equivalence identification. It preserves logical equivalence while leveraging LLM embeddings for enriched functionality. Extensive experiments validate MGVGA’s effectiveness, showing consistent improvements over DeepGate2 and competitive results against DeepGate3 for small-scale circuits. All reviewers except reviewer jhYv recommend acceptance. Considering that jhYv (score 5) did not engage into the discussion, a recommendation of acceptance is made.

**Additional Comments On Reviewer Discussion:**

Points Raised by Reviewers and Author Responses:
1. Scalability and Large-Scale Circuits (Raised by xqkr):
Concern: The scalability of MGVGA to handle large circuits was questioned, particularly regarding computational bottlenecks in masking and reconstruction.
Author Response: Authors highlighted the sparsity of digital circuits, the use of parallel processing and distributed computing, and their successful testing on large circuits (up to millions of gates). They also clarified that practical EDA workflows often involve graph partitioning, making ultra-large circuit handling less critical.

2. Baseline Comparisons (Raised by xqkr and jhYv): Concern: The absence of a comparison with DeepGate3 for small-scale circuits and HOGA for broader benchmarking was noted.
Author Response: The authors conducted additional experiments comparing MGVGA with DeepGate3 for small circuits, showing superior performance. They justified the omission of HOGA due to its computational inefficiency for large circuits and included further clarifications about training dataset alignment with practical needs.

3. Logical Equivalence and Terminology (Raised by EYP6): Concern: The concept of "logical equivalence" as used in the paper was questioned, as it seemed closer to "logical consistency" or "logical correctness."
Author Response: The authors clarified the intended meaning, distinguishing between logical equivalence in traditional circuit transformations and its use in their framework to denote consistent logical constraints in MGM and VGA. They proposed revising the terminology for clarity and updated the manuscript accordingly.

4. Additional Experiments on SAT Solving (Suggested by rsFP and EYP6): Concern: Practical downstream tasks like SAT solving were suggested as a more meaningful evaluation of MGVGA.
Author Response: The authors conducted SAT-solving experiments and showed significant performance improvements over baselines, demonstrating MGVGA's applicability to practical tasks.

Weighing of Points in the Final Decision:
The authors provided detailed, thoughtful, and actionable responses to all concerns, including new experiments, manuscript revisions, and clarifications. Their willingness to engage deeply with reviewer feedback improved the overall quality and clarity of the paper. They also added additional experiments addressing scalability, small-scale circuit comparisons, and SAT-solving tasks bolstered the evidence of MGVGA's effectiveness and practicality.

---

### Decision · Program_Chairs · 2025-01-22

Accept (Poster)